# Oligodendroglia-to-pericyte conversion after lipopolysaccharide exposure is gender-dependent

**Qingting Yu[1,2]**, **Linyuan Zhang[1]**, **Ting Xu[2]**, **Jiapeng Shao[2]**, **Falei Yuan[2]**, **Zuisu Yang[2]**, **Yuncheng Wu[1]***, **Haiyan Lyu[1]***

1 Department of Neurology, Shanghai General Hospital, Shanghai Jiao Tong University School of Medicine, Shanghai, China, 2 Department of Pharmacy, School of Food and Pharmacy, Zhejiang Ocean University, Zhoushan, China

☯ These authors contributed equally to this work.
* luhaiyan198109@163.com (HL); drwu2006@163.com (YW)

**Data Availability Statement:** The data is stored in the BioStudies, and the link is https://www.ebi.ac.uk/biostudies/studies/S-BSST1487.

**Funding:** This work was supported by the Natural Science Foundation of Shanghai (grant#

## Abstract

To investigate the sex-dependent differentiation of Sox10 cells and their response to pathological conditions such as lipopolysaccharide (LPS) exposure or ischemia, we utilized Sox10 Cre-ER[T2], tdTomato mice. Tamoxifen administration induced the expression of red fluorescent protein (RFP) in these cells, facilitating their subsequent tracking and analysis after LPS injection and ischemia via immunofluorescence staining. Propidium iodide (PI) was injected to label necrotic cells following LPS administration. We found that the conversion of Sox10 cells to pericytes in female mice was significantly higher than in male mice, especially in those exposed to LPS. After LPS injection, the number of PI[+] necrotic cells were significantly greater in females than in males. Moreover, RFP[+] cells did not co-localize with glial fibrillary acidic protein (GFAP) or cluster of differentiation 11b (CD11b). Similarly, after brain ischemia, RFP[+] cells did not express cluster of differentiation 13 (CD13), neuronal nuclei (NeuN), GFAP, or ionised calcium binding adaptor molecule 1 (Iba-1). These findings indicate that the conversion of Sox10 cells to pericytes following LPS exposure is sex-dependent, with neither male nor female groups showing differentiation into other cell types after LPS exposure or under ischemic conditions. The differences in LPS-induced necrosis of pericytes between sexes may explain the variations in the conversion of Sox10 cells to pericytes in both sexes.

## 1. Introduction

Sex differences significantly influence the prevalence and clinical outcomes of neurological disorders such as neurodegenerative diseases, stroke, and depression [1–3]. For instance, women are at a two-fold higher risk of developing Alzheimer's disease (AD) than men at the age of 45 [4]. Recent research has shown that myelin dysfunction occurs before amyloidosis, which is a characteristic hallmark of AD [5]. As a reservoir for oligodendrocytes (OLs), oligodendrocyte precursor cells (OPCs) continuously differentiate into OLs throughout adult life [6]. Sexual dimorphism has been reported in OPCs, with OPCs in female mice proliferating faster than male and the reason is still unknown [7]. OPCs have been reported to differentiate

21ZR1451700 to HYL) and the National Natural Science Foundation of China (grant# 82171286 to HYL and grant# 82101382 to FLY).

**Competing interests:** The authors declare that the research was conducted in the absence of any commercial or financial relationships that could be construed as a potential conflict of interest.

into pyramidal projection neurons within the mature piriform cortex, particularly producing glutamatergic neurons [8, 9]. Additionally, OPCs have also been shown to generated astrocytes [10, 11]. However, using a similar lineage tracing strategy, Kang et al. found that OPCs do not differentiate into neurons or astrocytes in young adult mice or in a mouse model of amyotrophic lateral sclerosis [12]. To our knowledge, the fate of OPCs in both sexes has not been studied using the Cre-loxP lineage tracing system in a neuroinflammation model.

Lipopolysaccharide (LPS) is a component of the outer membrane of Gram-negative bacteria and is often used to induce neuroinflammation. LPS has been found to co-localize with amyloid β in the brain parenchyma and vessels of AD patients [13]. LPS administration has been one of the nontransgenical model of AD [14]. LPS can also elicit noncanonical pyroptosis which is an important antibacterial defense against intracellular bacteria [15]. Research has shown that exposure to LPS results in a marked increase in pro-inflammatory markers in female mice compared to males [16]. Interestingly, while LPS has been found to impair learning abilities in male mice, it does not exhibit the same effect on females [17].

The blood-brain barrier (BBB) consists of pericytes, endothelial cells, and the endfeet of astrocytes. Along with adjacent microglia and OPCs, the BBB and neurons form the neurovascular unit (NVU), which is essential for maintaining homeostasis in the brain. Disruption of the BBB allows toxic molecules, cells, and microorganisms in the blood to enter the brain parenchyma, causing a strong inflammatory response and activating pathways related to neuronal death [18]. After an injury, pericytes are the first responder rather than microglia [19, 20]. For example, within one hour of cerebral ischemia, pericytes detach from the capillaries [21]. Research has shown that the BBB becomes leaky in the hippocampus of patients with mild cognitive impairment (MCI) [22]. Previously, we discovered that Sox10 cells can differentiate into pericytes to repair the injured BBB [23]. Sox10 serves as a marker within the oligodendrocyte lineage cells, encompassing OPCs, pre-myelinating, and myelinating oligodendrocytes [24]. The fate of Sox10 cells in both sexes under specific pathological conditions remains unclear. This study aimed to trace Sox10 cells after LPS-induced inflammation and cerebral ischemia in male and female mice.

## 2. Methods and materials

### 2.1 Animals

Male and female C57BL/6J mice were purchased from Hangzhou Ziyuan Experimental Animal Technology. Sox10 Cre-ER$^{T2}$ (#027651, Jackson Laboratory) and Rosa-tdTomato (#007909, Jackson Laboratory) mice were bred, and male/female offsprings from the F1 generation at 6 weeks of age were utilized for the induction of red fluorescent protein (RFP) with tamoxifen. Tamoxifen was administered orally in sunflower oil (1mg per mouse) for three consecutive days. The animal studies were approved by the Experimental Animal Ethics Committee of Zhejiang Ocean University (# SCXK ZHE 2019–0031).

### 2.2 Neuroinflammation

To initiate neuroinflammation, mice received a single intraperitoneal injection of 5 mg/kg LPS from Escherichia coli 055: B5 (#ST1470, Beyotime, Shanghai) or sterile saline, 14 days after the initial tamoxifen dose, and brain tissue was collected 7 days subsequently.

### 2.3 Distal middle cerebral artery occlusion (MCAO)

Fourteen days following the initial administration of tamoxifen, mice were anesthetized with 1.5% isoflurane mixture containing 70% nitrogen and 30% oxygen, delivered through a

vaporizer (RWD Life Science, Shenzhen, China). The left MCA was exposed and coagulated near the level of the brain fissure using an electric coagulator. Brain tissues were collected 7 days later after MCAO.

### 2.4 Immunofluorescence staining

All mice were euthanized with carbon dioxide and transcardially perfused with saline and 4% paraformaldehyde (PFA). Brain tissues were harvested, embedded in low-melting-point agarose sectioned by a vibratome (ZQP-86, Zhisun Equipment Inc., Shanghai) at a thickness of 50 μm. Sections were treated with 0.3% Triton X-100 and 1% bovine serum albumin in a 24-well cell culture plate. For immunofluorescence staining, the sections underwent overnight incubation at 4˚C with primary antibodies that target specific proteins: cluster of differentiation 13 (CD13, GTX75927, Genetex), platelet-derived growth factor receptor β (PDGFRβ, AF1042, R&D Systems; 14-1402-82, ThermoFisher), neuronal nuclei (NeuN, 266004, Synaptic Systems), GFAP (D262817, Sangon Biotech), cluster of differentiation 11b (CD11b, 557394, BD Bio-sciences), and ionized calcium-binding adapter molecule 1 (Iba-1, 019–19741, WAKO). The above antibodies were diluted to 1:400. Following three PBS washes, secondary antibodies (112-095-003, 111-095-003, 706-095-148, 705-095-147 Jackson ImmunoResearch) were applied at room temperature for 1 h. Subsequently, the nuclei were stained using 4',6-dia-midino-2-phenylindole (DAPI), and the sections were examined using a fluorescence micro-scope (Olympus BX41).

### 2.5 PI labeling of necrotic cells

Twenty-four hours after injecting LPS, PI (10 mg/kg, Aladdin, Shanghai) dissolved in saline was administered via the tail vein. Ten minutes later, the mice were perfused transcardially with saline, followed by 4% PFA. Brain tissues were then collected and sectioned as described in Section 2.4.

### 2.6 Statistical analysis

Statistical analysis and graph creation were carried out using GraphPad Prism 9.0 (GraphPad Software). The data were presented as mean ± standard error of the mean (SEM). The significance analysis was conducted using an unpaired t-test. $P < 0.05$ was deemed statistically significant.

## 3. Results

### 3.1 Females have more conversion of Sox10-A cells to pericytes than males with low-dose tamoxifen

In a previous study, we identified a population of pericyte precursor cells known as Sox10-A cells, which can differentiate into pericytes in adult mice [23]. To explore the influence of sex on this differentiation process, we analyzed the distribution of RFP+ cells to elucidate the disparities between male and female mice in the conversion of Sox10-A cells into pericytes. The number of Sox10-A cells was significantly higher in the cortex in the female group (2.2% ± 0.2%) than in the male group (1.1% ± 0.1%) (***$p < 0.001$, Fig 1A–1C). Immunofluorecent staining of CD13 and PDGFRβ showed that the number of RFP+ pericytes was significantly higher in the cortex in the female group (9.3% ± 0.6% CD13+ RFP+ pericytes, 15.5% ± 0.9% PDGFRβ+ RFP+ pericytes) than in the male group (3.8% ± 0.4% CD13+ RFP+ pericytes and 7.5% ± 0.7% PDGFRβ+ RFP+ pericytes) (***$p < 0.001$, Fig 1A, 1B, 1D and 1E). They also appeared in the thalamus of the female group (0.4% ± 0.1% Sox10-A cells, 1.6% ± 0.3% CD13+

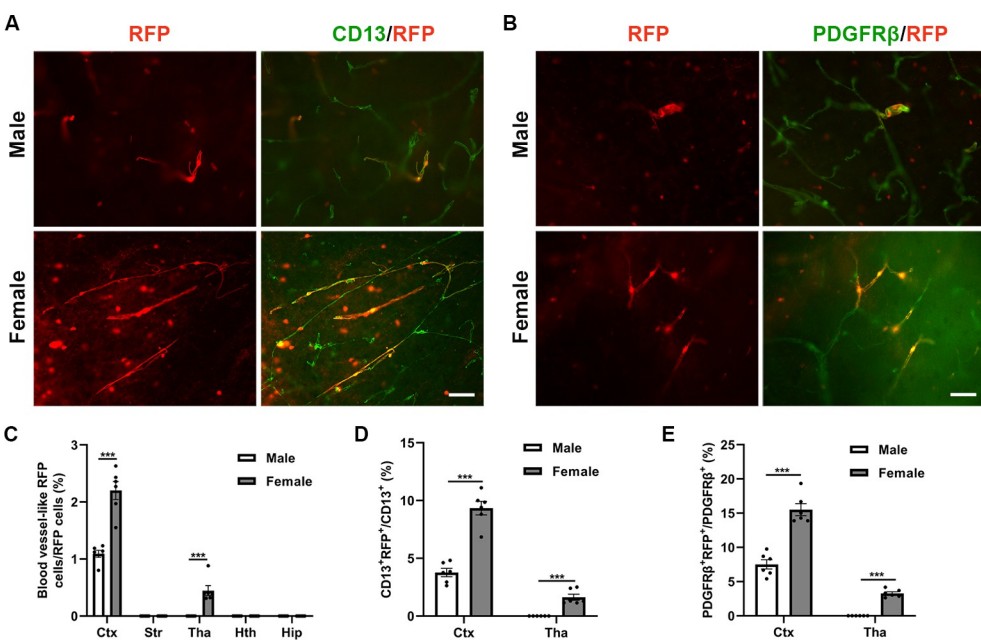

**Fig 1. Conversion of Sox10 cells to pericytes in different groups.** (A) Blood vessel-like RFP cells and immunofluorescence staining of CD13 and PDGFRβ in the cortex of both the male and female groups. (B) PDGFRβ immunostaining in the different groups. (C-E) Proportions of blood vessel-like RFP cells, CD13$^+$ RFP cells, and PDGFRβ$^+$ RFP$^+$ cells in the different groups. The data are presented as mean ± SEM. Scale bar: 20 μm. ***$p < 0.001$.

RFP$^+$ pericytes, 3.3% ± 0.2% PDGFRβ$^+$ RFP$^+$ pericytes) (***$p < 0.001$, Fig 1C–1E). These findings indicate that sex has a significant impact on the conversion of Sox10 cells to pericytes.

## 3.2 Females have more conversion of Sox10-A cells to pericytes than males after LPS administration

To assess the impact of LPS on the conversion of Sox10 cells to pericytes, we analyzed the effects in both male and female mice following LPS exposure. Seven days after a single dose of intraperitoneal injection of LPS, the number of Sox10-A cells was significantly higher in the cortex in the female group (5.4% ± 0.5%) than in the male group (1.0% ± 0.1%) (***$p < 0.001$, Fig 2A–2C). Immunofluorecent staining of CD13 and PDGFRβ showed that the number of RFP$^+$ pericytes was significantly higher in the cortex in the female group (35.1% ± 2.0% CD13$^+$ RFP$^+$ pericytes, 46.9% ± 2.1% PDGFRβ$^+$ RFP$^+$ pericytes) than in the male group (4.2% ± 0.3% CD13$^+$ RFP$^+$ pericytes, 9.0% ± 0.8% PDGFRβ$^+$ RFP$^+$ pericytes) (***$p < 0.001$, Fig 2A, 2B, 2D and 2E). Additionally, the converted Sox10-A cells also appeared in the striatum of the female group (1.5% ± 0.1% Sox10-A cells, 5.4% ± 0.6% CD13$^+$ RFP$^+$ pericytes, 9.5% ± 1.4% PDGFRβ$^+$ RFP$^+$ pericytes) (***$p < 0.001$, Fig 2C–2E). These results indicate that neuroinflammation plays a significant role in the conversion of Sox10 cells to pericytes in female mice.

## 3.3 Sox10-B cells do not convert into neurons after LPS administration

Previously, we identified a population of Sox10-B cells capable of differentiating into neurons in the cortex and the DG, and this process is linked to the toxicity of tamoxifen [23]. To investigate whether LPS affects the conversion of Sox10 cells to neurons, we conducted immunostaining for NeuN. Our results showed a lack of RFP$^+$ neurons in the cortex and DG across both male and female groups, including those subjected to LPS exposure (Fig 3). This absence

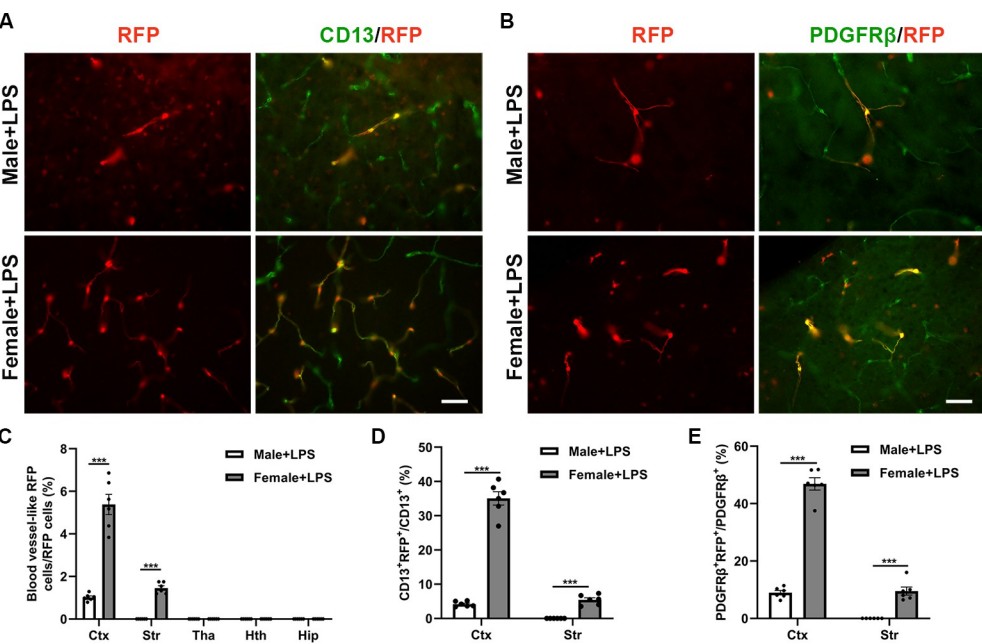

**Fig 2. Effect of LPS injection on the conversion of Sox10 cells to pericytes.** (A) Blood vessel-like RFP cells and immunofluorescence staining of CD13 in the cortex of the male + LPS group and the female + LPS group. (B) PDGFRβ immunostaining in the different groups. (C-E) Proportions of blood vessel-like RFP cells, CD13$^+$ RFP$^+$ cells and PDGFRβ$^+$ RFP$^+$ cells in the different groups. The data are presented as mean ± SEM. Scale bar: 20 μm. ***$p < 0.001$.

of conversion suggests that Sox10-B cells do not transform into neurons following LPS exposure, indicating that the conversion of Sox10 cells to neurons is not associated with neuroinflammation.

## 3.4 Sox10 cells do not differentiation into astrocytes or microglia after LPS administration

To examine whethe the Sox10 cells can differentiate into astrocytes and microglia, we performed immunostaining with GFAP and CD11b. We found that RFP$^+$ cells did not co-localize

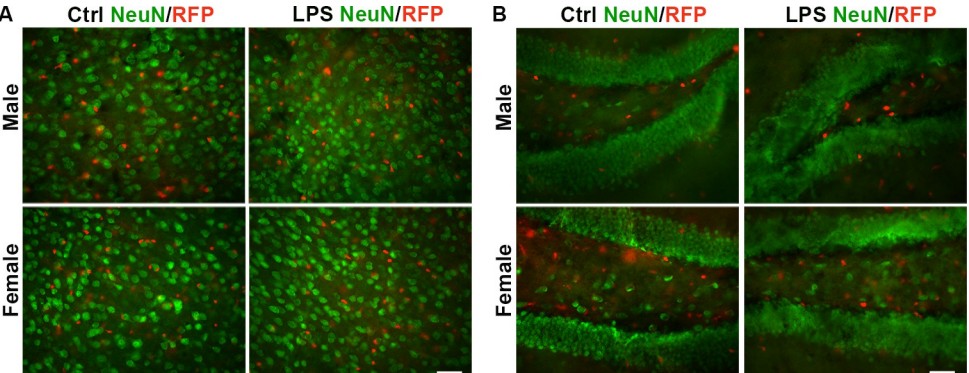

**Fig 3. Impact of LPS injection on the conversion of Sox10 cells to neurons.** (A) NeuN immunostaining in layer 2/3 of the cortex of male mice, female mice, male and female mice after LPS administration. (B) NeuN immunofluorescence staining in the DG across the groups. Scale bar: 20 μm.

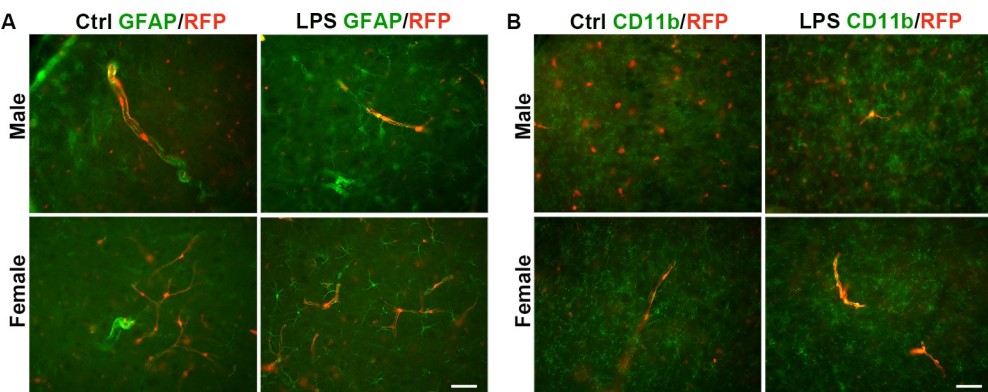

**Fig 4. RFP⁺ cells did not convert into astrocytes or microglia after LPS administration.** (A) Immunostaining for GFAP was performed in the cortex of the male group, the female group, the male + LPS group and the female + LPS group. (B) Immunostaining for CD11b was conducted in the cortex of across the groups. Scale bar: 20 μm.

with GFAP or CD11b in either male or female mice (Fig 4A and 4B). Likewise, seven days after intraperitoneal injection of LPS, RFP⁺ cells did not co-localize with GFAP or CD11b in either male or female mice (Fig 4A and 4B). Therefore, Sox10 cells do not differentiate into astrocytes or microglia following LPS exposure.

### 3.5 Pericytes in females are more susceptible to undergo necrosis after acute LPS administration

To understand the increased conversion of Sox10 cells in female mice, we injected PI through the tail vein after LPS administration, as PI labels necrotic cells after injury [25]. First, we tried the above described dose of LPS. PI⁺ cells (8.0% ± 0.3%) were only found in females 24 hours after injecting 5 mg/kg LPS (****$p < 0.0001$, Fig 5A). Next, we increased the dose of LPS and found that with 25 mg/kg LPS, although 8.7% ± 0.3% PI⁺ cells were in males, females still had significantly higher numbers of PI⁺ cells (64.6% ± 0.7%) (****$p < 0.0001$, Fig 5B). Among them, females had a significantly higher number of PI⁺ CD13⁺ pericytes (14.0% ± 0.2%) compared to males (7.3% ± 0.6%) (****$p < 0.0001$, Fig 5C). These results indicate that acute LPS administration leads to a higher rate of pericyte necrosis in females than males.

### 3.6 Ischemic stroke does not cause the differentiation of Sox10 cells

To determine the differentiation potential of RFP⁺ cells following ischemic stroke, we performed distal MCAO surgery on both male and female mice. Immnostaining results showed that RFP⁺ cells did not express CD13, NeuN, GFAP, or Iba-1 in either the male and female groups 7 days after MCAO (Fig 6A–6D). These results indicate that cerebral ischemia does not induce the differentiation of Sox10 cells into pericytes, neurons, astrocytes, or microglia in either male or female mice.

## 4. Discussion

In this study, we found that sex significantly influenced the remodeling of BBB after LPS administration. Specifically, pericytes in females were more susceptible to LPS challenge, potentially due to pericyte necrosis. Additionally, Sox10 cells did not convert into astrocytes or microglia following LPS administration. Moreover, after MCAO, Sox10 cells remained unchanged and did not differentiate into pericytes or neurons.

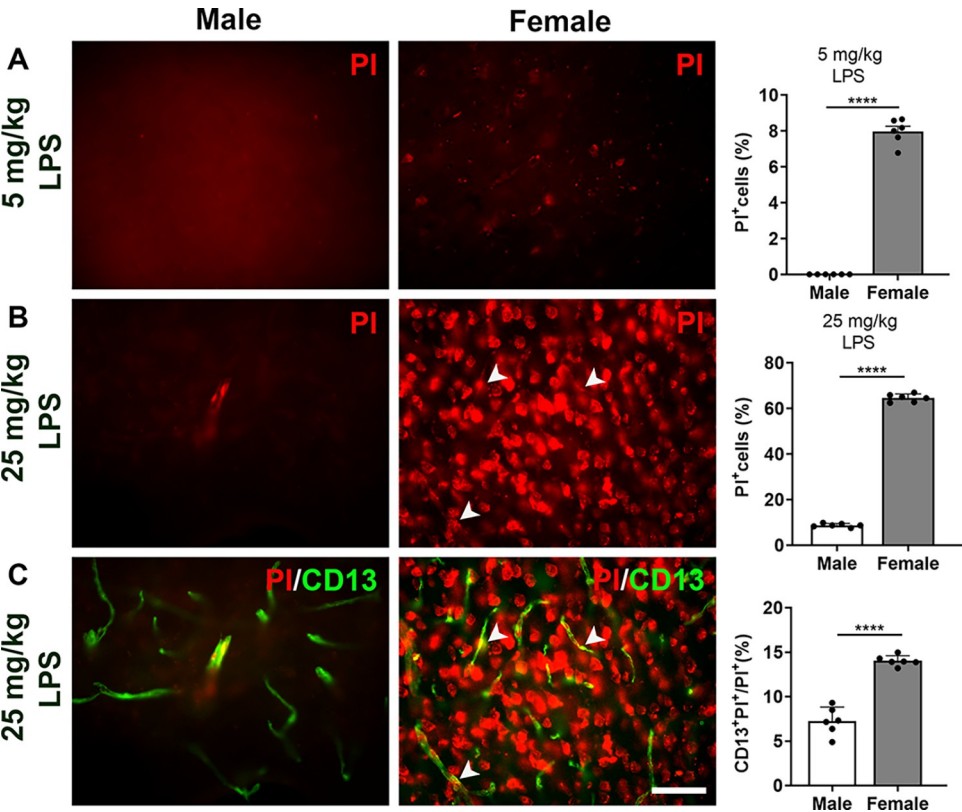

**Fig 5. Sex differences in LPS-induced BBB disruption in the mouse cortex.** (A) Images and cell percentages of PI-labeled cells in the cortex of males and females 24 hours after administering 5 mg/kg LPS. (B) Images and cell percentages of PI-labeled cells in the cortex of males and females 24 hours after administering 25 mg/kg LPS. (C) Images and cell percentages showing the co-localization of PI and CD13 in the cortex of males and females 24 hours after administering 25 mg/kg LPS. The data are presented as mean ± SEM. Scale bar: 20 μm. ****$p < 0.0001$.

Sox10 cells did not convert into neurons after LPS administration, possibly because a low dose of tamoxifen was used in this study. The conversion may be influenced by the extent of neuronal injury, which can be induced by higher doses of tamoxifen [23]. This finding

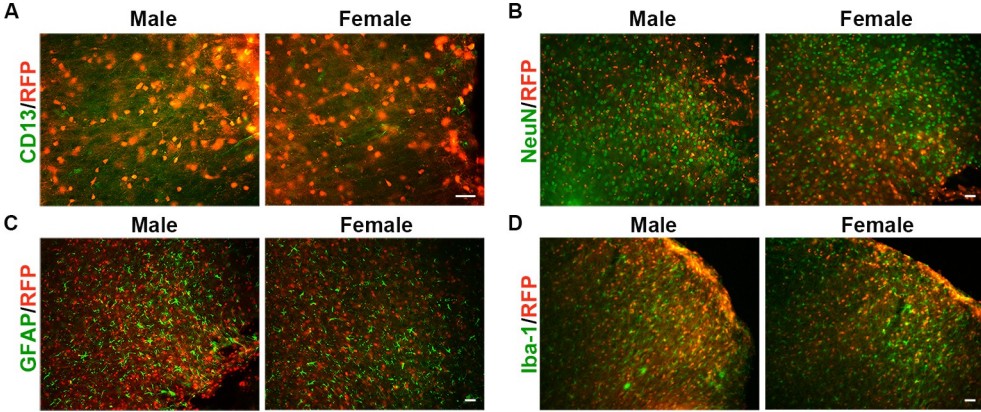

**Fig 6. Effect of cerebral ischemia on the differentiation of Sox10 cells.** (A-D) Immunostaining for CD13, NeuN, GFAP, and Iba-1 in the cortex of the ischemic group across different genders. Scale bar: 20 μm for A, 40 μm for B-D.

contrasts with the research conducted by Mayrhofer et al., who observed a significant increase in the transfer of cell material from Sox10 cells to neurons in the LPS-treated group compared to the control group [26]. Therefore, we suggest using tamoxifen-injected Cre mice of the same gender as a control. In addition to its widespread use in Cre-mediated lineage tracing, tamoxifen, a "selective estrogen receptor modulator" (SERM), is used for treating breast cancer in women. Tamoxifen is known to cause neuronal stress and cognitive decline, commonly called "TAM brain fog", a phenomenon that has not been well studied [27]. Furthermore, we cannot rule out the possibility that the disparity in the conversion of Sox10 cells to pericytes in both sexes is due to the effect of tamoxifen on the estrogen receptors in Sox10 cells, as oligodendrocyte lineage cells have been reported to have estrogen receptors such as ERα and Erβ [28]. Tamoxifen has also been demonstrated to inhibit the key enzyme of cholesterol metabolism [27, 29]. Dysfunction of cholesterol metabolism is an important risk factor for developing AD, and controlling cholesterol levels may provide a therapeutic effect for AD [30]. Therefore, the exact effect of tamoxifen on the central nervous system needs to be thoroughly studied in the future.

As shown by PI labeling results after LPS administration, females exhibit more necrotic pericytes than males when given the same dosage of LPS. This may trigger the differentiation of Sox10 cells to repair the injured BBB. Exposure to LPS may cause pericytes loss, which is crucial for understanding BBB disruption in inflammatory conditions [31, 32]. LPS promotes the formation of amyloid β, it would be interesting to check the converted Sox10 cells by LPS injection is related to the formation of amyloid β [33]. Retinal pericytes loss in AD patients has been attributed to apoptosis [34]. However, in a sepsis model, apoptosis is not involved in pericyte loss, suggesting alternative pathways of cell death [31]. Pyroptosis of pericytes emerges as a potential mechanism, especially in the context of sepsis [35]. Recent reserch has indicated that LPS-induced BBB breakdown is caused by endothelial pyroptosis but not pericyte pyroptosis [36]. However, the exact sex of the mice used in each experiment is not specified. Further studies are needed to determine whether the LPS dosage influences the analysis of inflammatory BBB damage, the role of pericytes in BBB breakdown, and the type of necrosis involved.

Our findings indicate that Sox10 cells do not differentiate into pericytes, neurons, astrocytes, or microglia following brain ischemia. Ischemia damages a large number of interpericyte tunnelling nanotubes (IP-TNTs) in the retina, which connect two pericytes [37]. This damage may explain why Sox10 cells cannot transform into pericytes after permanent MCAO. However, these IP-TNTs have been shown to regrow after transient ischemia [38]. Future studies should focus on the effects of transient ischemia, with various occlusion times, on the eliminiation of IP-TNTs and how to maintain these cell tunnels in stroke.

## Supporting information

**S1 Data.**
(XLSX)

## Author Contributions

**Conceptualization:** Falei Yuan, Yuncheng Wu, Haiyan Lyu.

**Data curation:** Qingting Yu, Linyuan Zhang.

**Formal analysis:** Qingting Yu, Falei Yuan, Yuncheng Wu, Haiyan Lyu.

**Funding acquisition:** Falei Yuan, Haiyan Lyu.

**Investigation:** Qingting Yu, Linyuan Zhang, Ting Xu, Jiapeng Shao, Zuisu Yang, Yuncheng Wu, Haiyan Lyu.

**Methodology:** Qingting Yu, Linyuan Zhang, Haiyan Lyu.

**Writing – original draft:** Qingting Yu, Linyuan Zhang, Haiyan Lyu.

**Writing – review & editing:** Yuncheng Wu, Haiyan Lyu.

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
