## [Decision Letter · Decision Letter 0]

24 May 2024

PONE-D-24-10180Gender-specific oligodendroglia-to-pericyte conversion following lipopolysaccharide administrationPLOS ONE

Dear Dr. Lyu,

Thank you for submitting your manuscript to PLOS ONE. After careful consideration, we feel that it has merit but does not fully meet PLOS ONE’s publication criteria as it currently stands. Therefore, we invite you to submit a revised version of the manuscript that addresses the points raised during the review process.

We look forward to receiving your revised manuscript.

Kind regards,

Syed M. Faisal, Ph.D.

Academic Editor

PLOS ONE

Journal Requirements:

"This work was supported by the Natural Science Foundation of Shanghai (grant# 21ZR1451700 to HYL) and the National Natural Science Foundation of China (grant# 82171286 to HYL and grant# 82101382 to FLY)."

3. Please upload a copy of Supporting Information Figure/Table/etc. "S1 Data. Raw data of this article" which you refer to in your text on page 23 in PDF submission.

**Additional Editor Comments:**

**If you disagree with any of the reviewer's comments, please provide a detailed rationale for each point of contention.**

**1. Clearly explain the observed disparity in conversion rates between sexes on exposure to LPS and address the reviewer's concern regarding the loss of pericytes.**

**2. Please revise the introduction section of the manuscript in light of the recent and most up to date references.**

Reviewers' comments:

Reviewer's Responses to Questions

**Comments to the Author**

1. Is the manuscript technically sound, and do the data support the conclusions?

Reviewer #1: No

Reviewer #2: Yes

2. Has the statistical analysis been performed appropriately and rigorously? 

Reviewer #1: Yes

Reviewer #2: Yes

3. Have the authors made all data underlying the findings in their manuscript fully available?

Reviewer #1: Yes

Reviewer #2: Yes

4. Is the manuscript presented in an intelligible fashion and written in standard English?

Reviewer #1: No

Reviewer #2: Yes

5. Review Comments to the Author

Reviewer #1: The research article entitled “Gender-specific oligodendroglia-to-pericyte conversion following lipopolysaccharide administration” is interesting in terms of its novelty in exploring the differentiation of oligodendroglia in different sexes and pathological conditions. However, the article has major shortcomings in the presentation. It lacks the proper flow and clear explanations.

• The introduction is poorly written. The authors have mentioned many sentences that are not necessary and, on the contrary, didn’t include the information that is required in the manuscript. In the abstract section authors have mentioned that conversion of female Sox10 cells to pericytes was significantly higher than that of males when exposed to LPS. In the following sentence, they mention the conversion rate of RFP cells is higher in the untreated female and the LPS-treated male group, with the latter not significantly different from the untreated male group. The authors have not mentioned the difference between the sox10 cells and RFP cells and have used them interchangeably. If they meant it to be the same, why are the authors mentioning the conversion rate of RFP cells is higher in the LPS-treated male group? It doesn't need to be mentioned when it's significantly not different from the untreated male group. Also, is it so important to mention it in the abstract, and wherein the main data they have shown it?

• In the results section 3.1, what does low-dose condition mean? It’s not clearly mentioned. The heading says females have more sox10A cells while it should be “females have more conversion of sox10A to pericytes”.

• Support lines 252-255 with references.

• Authors have shown the conversion of Sox10 cells to pericyte on exposure to LPS which is significantly higher in females compared to males. In the discussion section, the authors mention the loss of pericyte on exposure to LPS. Authors should clearly explain this disparity/confusion in the discussion section.

• Overall, the manuscript needs rewriting to present the data explaining every relevant thing clearly and explicitly in detail before further consideration.

Reviewer #2: The study explores sex-specific differences in the conversion of Sox10 cells into pericytes and its implications for blood-brain barrier integrity. Female mice show enhanced differentiation compared to males. The findings highlight the importance of considering sex-specific responses in addressing blood-brain barrier disruptions in neurodegenerative conditions.

1. Could the authors provide more insight into the mechanisms underlying the observed sex-specific differences in the conversion of Sox10 cells into pericytes, particularly regarding the role of Erα and potential interactions with other signaling pathways?

2. In light of the contrasting findings with previous studies on Sox10 cell differentiation into neurons and astrocytes, what factors do the authors believe might account for these discrepancies, and how could future research reconcile these differences?

3. Can the authors elaborate on the potential clinical implications of their findings regarding the influence of low-dose tamoxifen treatment on Sox10 cell differentiation pathways, particularly in the context of enhancing blood-brain barrier repair and slowing the progression of neurodegenerative diseases?

4. Considering the limitations mentioned in the article, such as potential confounding factors, could the authors discuss how these limitations might have affected the interpretation of the results and suggest strategies for addressing them in future studies?

5. How do the findings of this study contribute to our understanding of pericyte loss in neurodegenerative diseases like Alzheimer's disease and stroke, and what avenues for further research do the authors believe should be explored to build upon these findings?

6. Are there any specific experimental approaches or analyses that the authors believe would strengthen the conclusions of the study or provide additional insights into the mechanisms underlying sex-specific responses in Sox10 cell differentiation?

6. PLOS authors have the option to publish the peer review history of their article (what does this mean?). If published, this will include your full peer review and any attached files.

Reviewer #1: No

Reviewer #2: No

---

## [Author Response · Author response to Decision Letter 0]

12 Jul 2024

Editor Comments:

1. Please note that funding information should not appear in any section or other areas of your manuscript. We will only publish funding information present in the Funding Statement section of the online submission form. Please remove any funding-related text from the manuscript.

Ans：We have removed funding-related text from the manuscript.

Ans：We have re-uploaded the data to the BioStudies. The link is https://www.ebi.ac.uk/biostudies/studies/S-BSST1487

Editor Comments:

Ans: We have adjusted the formatting according to the guidelines.

2. Please state what role the funders took in the study. If the funders had no role, please state: ""The funders had no role in study design, data collection and analysis, decision to publish, or preparation of the manuscript."" If this statement is not correct you must amend it as needed. Please include this amended Role of Funder statement in your cover letter; we will change the online submission form on your behalf.

Ans: We have added the role of funder statement in the cover letter.

3. Please upload a copy of Supporting Information Figure/Table/etc. "S1 Data. Raw data of this article" which you refer to in your text on page 23 in PDF submission.

Ans: We have uploaded the “S1_Data sheet (XLSX)” of this article” and the raw data is stored in the BioStudies. 

The link is https://www.ebi.ac.uk/biostudies/studies/S-BSST1487

Ans: We have removed the reference: Montagne A, Nikolakopoulou AM, Zhao Z, Sagare AP, Si G, Lazic D, et al. Pericyte degeneration causes white matter dysfunction in the mouse central nervous system. Nat Med. 2018;24(3):326-37. doi: 10.1038/nm.4482.

Additional Editor Comments:

Reviewer #1: The research article entitled “Gender-specific oligodendroglia-to-pericyte conversion following lipopolysaccharide administration” is interesting in terms of its novelty in exploring the differentiation of oligodendroglia in different sexes and pathological conditions.

1. However, the article has major shortcomings in the presentation. It lacks the proper flow and clear explanations. The introduction is poorly written. The authors have mentioned many sentences that are not necessary and, on the contrary, didn’t include the information that is required in the manuscript. 

Ans：Thank you very much for the suggestion. We have revised and re-written the manuscript to make it clear and straightforward.

2. In the abstract section authors have mentioned that conversion of female Sox10 cells to pericytes was significantly higher than that of males when exposed to LPS. In the following sentence, they mention the conversion rate of RFP cells is higher in the untreated female and the LPS-treated male group, with the latter not significantly different from the untreated male group. The authors have not mentioned the difference between the Sox10 cells and RFP cells and have used them interchangeably. If they meant it to be the same, why are the authors mentioning the conversion rate of RFP cells is higher in the LPS-treated male group? It doesn't need to be mentioned when it's significantly not different from the untreated male group. Also, is it so important to mention it in the abstract, and where in the main data they have shown it?

Ans：Thank you for the helpful suggestion. We have deleted the incorrect sentences and revised the manuscript. Please see the revision.

3. In the results section 3.1, what does low-dose condition mean? It’s not clearly mentioned. The heading says females have more Sox10-A cells while it should be “females have more conversion of Sox10-A to pericytes”.

Ans：We have corrected this in the manuscript.

4. Support lines 252-255 with references.

Ans: We have removed the paragraphs to enhance the clarity of the manuscript and discussed it in the revision.

5. Authors have shown the conversion of Sox10 cells to pericyte on exposure to LPS which is significantly higher in females compared to males. In the discussion section, the authors mention the loss of pericyte on exposure to LPS. Authors should clearly explain this disparity/confusion in the discussion section.

Ans: We have added new data on propidium Iodide labeling of necrotic cells to the article as Fig.5.

6. Overall, the manuscript needs rewriting to present the data explaining every relevant thing clearly and explicitly in detail before further consideration.

Ans: We have re-written the manuscript according to the reviewer’s suggestion.

Reviewer #2: The study explores sex-specific differences in the conversion of Sox10 cells into pericytes and its implications for blood-brain barrier integrity. Female mice show enhanced differentiation compared to males. The findings highlight the importance of considering sex-specific responses in addressing blood-brain barrier disruptions in neurodegenerative conditions.

1. Could the authors provide more insight into the mechanisms underlying the observed sex-specific differences in the conversion of Sox10 cells into pericytes, particularly regarding the role of Erα and potential interactions with other signaling pathways?

Ans: We have discussed the potential interactions with cholesterol metabolism in the article. Please see the discussion.

2. In light of the contrasting findings with previous studies on Sox10 cell differentiation into neurons and astrocytes, what factors do the authors believe might account for these discrepancies, and how could future research reconcile these differences?

Ans: We think high doses of tamoxifen may cause neuronal injury and further induce the conversion of Sox10 cells to neurons. We have discussed this in the revision. Regarding previous studies on the conversion of OPCs to astrocytes, we believe more biomarkers like ALDH1L1, Sox9, and AQP4 should be used to confirm astrocytes instead of relying solely on GFAP.

3. Can the authors elaborate on the potential clinical implications of their findings regarding the influence of low-dose tamoxifen treatment on Sox10 cell differentiation pathways, particularly in the context of enhancing blood-brain barrier repair and slowing the progression of neurodegenerative diseases?

Ans: We have discussed this in the revision.

4. Considering the limitations mentioned in the article, such as potential confounding factors, could the authors discuss how these limitations might have affected the interpretation of the results and suggest strategies for addressing them in future studies?

Ans: We suggest using tamoxifen-injected Cre mice of the same gender as controls, rather than just wild-type controls. We have discussed it in the manuscript.

5. How do the findings of this study contribute to our understanding of pericyte loss in neurodegenerative diseases like Alzheimer's disease and stroke, and what avenues for further research do the authors believe should be explored to build upon these findings?

Ans: Pericyte loss may be related to the formation of amyloid β in Alzheimer's disease and disruption of interpericyte tunnelling nanotubes (IP-TNTs) in stroke. We have discussed it in the revision.

6. Are there any specific experimental approaches or analyses that the authors believe would strengthen the conclusions of the study or provide additional insights into the mechanisms underlying sex-specific responses in Sox10 cell differentiation?

Ans: Thank you for your suggestion. We have added new data on propidium Iodide labeling of necrotic cells after LPS injection to the article, presented as Fig.5.

---

## [Editor Report · Decision Letter 1]

18 Jul 2024

Oligodendroglia-to-pericyte conversion after lipopolysaccharide exposure is gender-dependent.

PONE-D-24-10180R1

Dear Dr. Lyu,

We’re pleased to inform you that your manuscript has been judged scientifically suitable for publication and will be formally accepted for publication once it meets all outstanding technical requirements.

Kind regards,

Syed M. Faisal, Ph.D.

Academic Editor

PLOS ONE
---

## [Editor Report · Acceptance letter]

26 Jul 2024

PONE-D-24-10180R1 

PLOS ONE

Dear Dr. Lyu, 

I'm pleased to inform you that your manuscript has been deemed suitable for publication in PLOS ONE. Congratulations! Your manuscript is now being handed over to our production team.

Kind regards, 

on behalf of

Dr. Syed M. Faisal 

Academic Editor

PLOS ONE